

# Genetic and morphological analyses indicate that the Australian endemic scorpion *Urodacus yaschenkoi* (Scorpiones: Urodacidae) is a species complex

Karen Luna-Ramirez[1], Adam D. Miller[3] and Gordana Rašić[2]

[1] Museum Victoria, Melbourne, Victoria, Australia
[2] Melbourne, Victoria, Australia
[3] Centre for Integrative Ecology, School of Life and Environmental Sciences, Deakin University, Victoria, Australia

## ABSTRACT

**Background**. Australian scorpions have received far less attention from researchers than their overseas counterparts. Here we provide the first insight into the molecular variation and evolutionary history of the endemic Australian scorpion *Urodacus yaschenkoi*. Also known as the inland robust scorpion, it is widely distributed throughout arid zones of the continent and is emerging as a model organism in biomedical research due to the chemical nature of its venom.

**Methods**. We employed Bayesian Inference (BI) methods for the phylogenetic reconstructions and divergence dating among lineages, using unique haplotype sequences from two mitochondrial loci (*COXI*, *16S*) and one nuclear locus (*28S*). We also implemented two DNA taxonomy approaches (GMYC and PTP/dPTP) to evaluate the presence of cryptic species. Linear Discriminant Analysis was used to test whether the linear combination of 21 variables (ratios of morphological measurements) can predict individual's membership to a putative species.

**Results**. Genetic and morphological data suggest that *U. yaschenkoi* is a species complex. High statistical support for the monophyly of several divergent lineages was found both at the mitochondrial loci and at a nuclear locus. The extent of mitochondrial divergence between these lineages exceeds estimates of interspecific divergence reported for other scorpion groups. The GMYC model and the PTP/bPTP approach identified major lineages and several sub-lineages as putative species. Ratios of several traits that approximate body shape had a strong predictive power (83–100%) in discriminating two major molecular lineages. A time-calibrated phylogeny dates the early divergence at the onset of continental-wide aridification in late Miocene and Pliocene, with finer-scale phylogeographic patterns emerging during the Pleistocene. This structuring dynamics is congruent with the diversification history of other fauna of the Australian arid zones.

**Discussion**. Our results indicate that the taxonomic status of *U. yaschenkoi* requires revision, and we provide recommendations for such future efforts. A complex evolutionary history and extensive diversity highlights the importance of conserving *U. yaschenkoi* populations from different Australian arid zones in order to preserve patterns of endemism and evolutionary potential.

Corresponding author
Gordana Rašić,
rasic.gordana@gmail.com

## INTRODUCTION

Scorpions represent an ancient arthropod lineage that first appeared in the Silurian, and fossil records indicate their bodyplan remained largely unchanged since the Paleozoic period (*Dunlop, 2010*; *Jeram, 1997*; *Kjellesvig-Waering, 1986*). Given this relative morphological stasis over long periods of time, the placement of scorpions within Arachnida and internal evolutionary relationships inferred solely from morphological characters have long been contentious (*Prendini & Wheeler, 2005*; *Sharma et al., 2014*; *Shultz, 2007*; *Soleglad & Fet, 2003*). A recent phylogenomic study based on the transcriptome-wide variation suggested non-monophyly of all scorpion superfamilies and several families, largely contradicting the traditional morphology-based hypotheses (*Sharma et al., 2015*).

The well-supported phylogenetic reconstructions and taxonomy of scorpions are critical for their effective conservation. Scorpion populations can be sensitive to environmental changes due to a low reproductive rate (long generation time, long gestation time, small litter size) and high mortality of immature females (*Fet, Polis & Sissom, 1998*; *Lourenço & Cuellar, 1995*). Several species have gained threatened status due to over-harvesting for the souvenir and exotic pet trades (*Convention on International Trade in Endangered Species of Wild Fauna and Flora, 2014*, http://www.cites.org/eng/app/appendices.php). Scorpions might also become more harvested for their venom that is increasingly regarded as a source of new therapeutic and insecticidal agents (*Gurevitz et al., 2007*; *Possani et al., 2000*; *Rodríguez de la Vega, Schwartz & Possani, 2010*). An extensive venom characterization can be found for individual taxa (e.g., *Luna-Ramírez et al., 2013*; *Xu et al., 2014*), but a deeper understanding of the evolution of scorpion venoms and their molecular characteristics has been limited by the lack of underlying species tree (*Sharma et al., 2015*).

Extant scorpions inhabit a diversity of terrestrial habitats across all continents except Antarctica, with the greatest species diversity found in tropical and subtropical regions of the world (*Lourenço, 2001*; *Prendini, 2010*). Australian scorpions have received far less attention from researchers than their overseas counterparts. Over 40 scorpion species described in Australia are traditionally organized into four families: Buthidae, Bothriuridae, Urodacidae and Hormuridae (*Koch, 1977*; *Monod & Prendini, 2015*; *Volschenk, Mattoni & Prendini, 2008*). The Urodacidae is an Australian endemic family found across the continent, except on the south-eastern seaboard. The family was first described by *Koch (1977)* that under the current classification includes two genera: *Urodacus* and the recently described troglobitic *Aops* (*Volschenk & Prendini, 2008*). The genus *Urodacus* contains 20 species described based on morphological characters (*Volschenk, Harvey & Prendini, 2012*), with many likely undescribed species.

*Urodacus yaschenkoi* (*Birula, 1903*), commonly known as the inland robust scorpion, occupies Australian desert habitats stretching from north-western Victoria through South Australia and across to Western Australia (*Walker, Yen & Milledge, 2003*) (Fig. 1). It is emerging as a model organism in toxinology because it produces large volumes of venom compared with other *Urodacus* species (*Luna-Ramírez et al., 2013*; *Luna-Ramírez et al., 2014*). This scorpion has had several synonyms throughout its taxonomic history, starting from the original description as *Hemihoplopus yaschenkoi* (*Birula, 1903*), followed

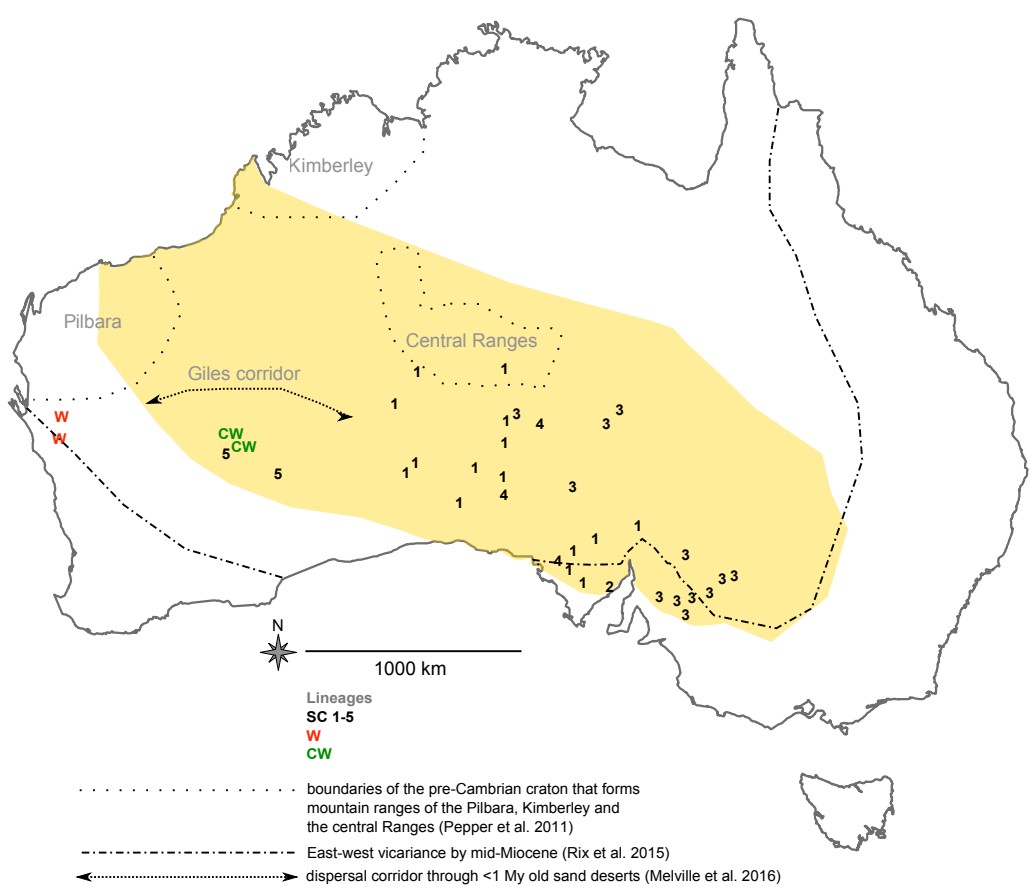

**Figure 1** *Urodacus yaschenkoi* **sampling locations across its distribution range (in dark yellow, adapted from** *Koch, 1977***).** Numbers 1 to 5 designate individuals belonging to the sub-lineages (SC1-5) of the south-central major clade (SC); members of the central-western (CW) clade and western (W) clades are marked in green and red color, respectively. Different hypotheses about diversification in various Australian taxa (vicariance, refugia, dispersal corridors) are adapted from *Melville et al. (2016)*, *Pepper et al. (2011)* and *Rix et al. (2015)*.

by *Urodacus granifrons* (*Kraepelin, 1916*), *U. fossor* (*Kraepelin, 1916*), and *U. kraepelini* (*Glauert, 1963*), and finally by *U. yaschenkoi* (Birula) (*Koch, 1977*). Since then, studies of variation in *U. yashenkoi* populations have not been conducted.

Here we provide the first molecular analysis of phylogenetic patterns and history of *U. yaschenkoi* sampled across its native range. DNA sequence data from mitochondrial and nuclear loci, complemented with the analysis of several body-proportion characters, showed that *U. yaschenkoi* shares a complex diversification history with other Australian arid-adapted fauna. Moreover, the existence of several deeply divergent lineages that also differ in body-shape indicate that further revision of this taxon is warranted.

## MATERIALS AND METHODS

### Biological material

Samples of *Urodacus yaschenkoi* were obtained from field and museum collections (Table 1). Live specimens were collected from eight locations (approximately 500 m$^2$) in the semi-arid
**Table 1** *Urodacus yaschenkoi* **specimen location and analyses made.** List of *Urodacus yaschenkoi* collected from the field as live specimens (Field) or obtained from the Australian museum collections (South Australian Museum—SA, Western Australian Museum—WA). Geographic position (lat/log) and the geographic region details are reported for each sample. List of haplotypes (mito *28S*) and GenBank Accession # scored in each individual. Morphological variation scored (✓), Museum ID.

| Sample | Source | Latitude | Longitude | Geographic region | Mito haplotype | 28S haplotype | GenBank (mito/28S) | Morpho | Museum ID/Reg.No. |
|--------|--------|----------|-----------|-------------------|----------------|---------------|--------------------|--------|-------------------|
| BKA11 | Field | −33.2283 | 141.3011 | NSW | 20 | 1 | KP176775/ KP176743 | | NA |
| BKA12 | Field | −33.2283 | 141.3011 | NSW | 20 | 2 | KP176775/ KP176744 | | NA |
| BKB08 | Field | −33.2199 | 141.3089 | NSW | 20 | 1 | | | NA |
| BKB12 | Field | −33.2242 | 141.3061 | NSW | 20 | 1 | KP176775/ KP176743 | | NA |
| BK13 | Field | −33.2283 | 141.3011 | NSW | 20 | 1 | | | NA |
| MARR1 | Field | −26.3400 | 133.2000 | SA | 28 | 3 | KP176783/ KP176745 | | NA |
| MARR2 | Field | −26.3400 | 133.2000 | SA | 28 | 3 | KP176783/ KP176745 | | NA |
| PIM1 | Field | −31.2509 | 136.5089 | SA | 1 | 4 | KP176756/ KP176746 | | NA |
| PIM2 | Field | −31.2509 | 136.5089 | SA | 1 | 5 | KP176756/ KP176747 | | NA |
| PIM5 | Field | −31.2509 | 136.5089 | SA | 1 | 4 | KP176756/ KP176746 | | NA |
| PIM6 | Field | −31.2509 | 136.5089 | SA | 1 | 1 | KP176756/ KP176743 | | NA |
| PIM8 | Field | −31.2509 | 136.5089 | SA | 1 | 1 | KP176756/ KP176743 | | NA |
| POP1 | Field | −33.0710 | 141.6372 | NSW | 20 | 1 | KP176775/ KP176743 | | NA |
| POP4 | Field | −33.0710 | 141.6372 | NSW | 20 | – | KP176775 | | NA |
| POP5 | Field | −33.0710 | 141.6372 | NSW | 20 | – | KP176775 | | NA |
| SAM1397 | SAM | −30.7667 | 138.1767 | SA | 2 | – | KP176757 | ✓ | NS1397 |
| SAM1399 | SAM | −27.1192 | 132.8300 | SA | 6 | – | KP176761 | ✓ | NS1399 |
| SAM1400 | SAM | −27.1191 | 132.8300 | SA | 6 | – | KP176761 | ✓ | NS1400 |
| SAM1403 | SAM | −26.6453 | 132.8858 | SA | 4 | – | KP176759 | ✓ | NS1403 |
| SAM1406 | SAM | −31.2878 | 136.5831 | SA | 1 | – | KP176756 | ✓ | NS1406 |
| SAM1412 | SAM | −26.2747 | 137.3269 | SA | 20 | – | KP176775 | ✓ | NS1412 |
| SAM1415 | SAM | −33.8555 | 140.5361 | SA | 20 | – | KP176775 | ✓ | NS1415 |
| SAM1416 | SAM | −34.0583 | 140.1500 | SA | 20 | – | KP176775 | ✓ | NS1416 |
| SAM1606 | SAM | −26.6922 | 134.1722 | SA | 23 | – | KP176778 | | NS1606 |
| SAM1607 | SAM | −26.5767 | 137.1933 | SA | 22 | – | KP176777 | | NS1607 |
| SAM1812 | SAM | −33.3267 | 137.0931 | SA | 15 | – | KP176770 | ✓ | NS1812 |
| SAM1823 | SAM | −33.7511 | 140.2747 | SA | 20 | – | KP176775 | ✓ | NS1823 |
| SAM1825 | SAM | −33.7230 | 140.1238 | SA | 20 | – | KP176775 | ✓ | NS1825 |
| SAM1831 | SAM | −33.7183 | 139.9300 | SA | 20 | – | KP176775 | ✓ | NS1831 |

**Table 1** (*continued*)

| Sample | Source | Latitude | Longitude | Geographic region | Mito haplotype | 28S haplotype | GenBank (mito/28S) | Morpho | Museum ID/Reg.No. |
|--------|--------|----------|-----------|-------------------|----------------|---------------|--------------------|--------|-------------------|
| SAM1834 | SAM | −33.7236 | 139.0438 | SA | 20 | – | KP176775 | ✓ | NS1834 |
| SAM1835 | SAM | −33.7236 | 139.0438 | SA | 21 | – | KP176776 | ✓ | NS1835 |
| SAM1837 | SAM | −33.7400 | 139.0816 | SA | 20 | – | KP176775 | ✓ | NS1837 |
| SAM1917 | SAM | −32.6244 | 135.0322 | SA | 24 | – | KP176779 | ✓ | NS1917 |
| SAM1939 | SAM | −33.1233 | 136.0214 | SA | 3 | – | KP176758 | ✓ | NS1939 |
| SAM2038 | SAM | −33.1167 | 136.0000 | SA | 3 | – | KP176758 | ✓ | NS2038 |
| SAM2053 | SAM | −24.4036 | 132.8886 | NT | 14 | – | KP176769 | ✓ | NS2053 |
| SAM2054 | SAM | −28.4627 | 129.0102 | SA | 5 | – | KP176760 | ✓ | NS2054 |
| SAM2055 | SAM | −28.4627 | 129.0102 | SA | 5 | – | KP176770 | ✓ | NS2055 |
| SAM2056 | SAM | −28.4627 | 129.0102 | SA | 10 | – | KP176765 | ✓ | NS2056 |
| SAM2060 | SAM | −28.4977 | 129.3205 | SA | 11 | – | KP176766 | ✓ | NS2060 |
| SAM2061 | SAM | −28.4977 | 129.3205 | SA | 11 | – | KP176766 | ✓ | NS2061 |
| SAM2062 | SAM | −24.5060 | 129.2619 | NT | 9 | – | KP176764 | ✓ | NS2062 |
| SAM2067 | SAM | −32.0033 | 135.6558 | SA | 3 | – | KP176758 | | NS2067 |
| SAM2070 | SAM | −28.8969 | 132.7575 | SA | 12 | – | KP176767 | ✓ | NS2070 |
| SAM2071 | SAM | −28.8969 | 132.7575 | SA | 13 | – | KP176768 | ✓ | NS2071 |
| SAM2073 | SAM | −28.5319 | 131.6903 | SA | 19 | – | KP176774 | | NS2073 |
| SAM2076 | SAM | −29.7706 | 131.1081 | SA | 18 | – | KP176773 | | NS2076 |
| SAM2120 | SAM | −31.9972 | 140.0644 | SA | 20 | – | KP176775 | ✓ | NS2120 |
| SAM2125 | SAM | −29.1286 | 135.6997 | SA | 25 | – | KP176780 | ✓ | NS2125 |
| SAM2126 | SAM | −29.1286 | 135.6997 | SA | 20 | – | KP176775 | | NS2126 |
| SAM2133 | SAM | −32.4947 | 135.3644 | SA | 7 | – | KP176762 | ✓ | NS2133 |
| SAM2140 | SAM | −29.4053 | 132.8556 | SA | 26 | – | KP176781 | ✓ | NS2140 |
| WAM20 | WAM | −27.4867 | 122.3119 | WA | 31 | 7 | KP176786/ KP176749 | ✓ | 85020 |
| WAM31 | WAM | −27.4867 | 122.3119 | WA | 31 | 8 | KP176786/ KP176750 | ✓ | 85031 |
| WAM32 | WAM | −27.4867 | 122.3119 | WA | 30 | 8 | KP176785/ KP176750 | ✓ | 85032 |
| WAM36 | WAM | −27.3893 | 115.1847 | WA | 29 | 9 | KP176784/ KP176751 | | 78236 |
| WAM37 | WAM | −27.6145 | 121.9947 | WA | 17 | 10 | KP176772/ KP176752 | | 112637 |
| WAM38 | WAM | −26.4408 | 115.3661 | WA | 29 | 9 | KP176784/ KP176751 | | 78238 |
| WAM46 | WAM | −28.7333 | 123.8667 | WA | 16 | 11 | KP176771/ KP176753 | | 80246 |
| WAM55 | WAM | −27.4867 | 122.3119 | WA | 31 | 7 | KP176786/ KP176749 | ✓ | 83855 |
| WAM56 | WAM | −27.4867 | 122.3119 | WA | 30 | 7 | KP176785/ KP176749 | ✓ | 83856 |
| WAM75 | WAM | −27.4867 | 122.3119 | WA | 31 | 12 | KP176786/ KP176754 | ✓ | 83875 |

**Table 1** (*continued*)

| Sample | Source | Latitude | Longitude | Geographic region | Mito haplotype | 28S haplotype | GenBank (mito/28S) | Morpho | Museum ID/Reg.No. |
|--------|--------|----------|-----------|-------------------|----------------|---------------|--------------------|--------|--------------------|
| WAM88 | WAM | −25.9307 | 128.4526 | WA | 8 | 13 | KP176763/ KP176755 | | 95988 |
| Um1814 | SAM | −33.1997 | 138.2189 | SA | NA | NA | | | NS0001814 |
| Um2714 | SAM | −33.1997 | 138.2189 | SA | NA | NA | | | NS0002714 |
| Un2112 | SAM | −31.6597 | 129.1083 | SA | NA | NA | | | NS0002112 |

**Notes.**
NSW, New South Wales; SA, South Australia; WA, Western Australia; NT, Northern Territory; NA, Not applicable.

and arid regions of Central Australia in December 2010 and October 2011 (Table 1 and Fig. 1). Individuals were collected at night from pitfall traps set in front of their burrows, and those outside their burrows were detected using ultraviolet (UV) lamps that reveal soluble fluorescent components (*β*-carboniles) in the scorpion exoskeleton (*Stachel, Stockwell & Van Vranken, 1999*). Captured scorpions were kept alive and transported to the laboratory for morphological identification according to *Koch (1977)*. Key diagnostic feature that distinguishes *U. yaschenkoi* from other *Urodacus* species is a very small terminal prolateral tarsus unguis. All specimens were handled according to good animal practices defined by the Government of Australia, and all institutions and museums involved approved the animal handling work. Scorpions were anaesthetized by cooling in a refrigerator (4 °C) for 5 min before removing ∼1 mm$^2$ of leg muscle tissue, which was stored in 90% ethanol at 4 °C or −20 °C for subsequent DNA extraction. Additional samples were obtained from collections at the South Australian Museum (SAM) and Western Australian Museum (WAM) containing specimens collected between 2000 and 2010 (Table 1).

## DNA extraction, amplification and sequencing

Total DNA was extracted from the stored muscle tissue using the DNeasy Blood and Tissue Kit (Qiagen, Venlo, Netherlands) following the manufacturer's instructions. Two mitochondrial loci (cytochrome oxidase subunit I, *COXI*; large ribosomal subunit, *16S*) and a single nuclear locus (*28S*) were amplified by PCR with a reaction volume of 20 μl containing 0.5 ng of template DNA, 10 μl of Go Taq Master Mix (Promega, Madison, Wisconsin, USA), 0.5 μl of 10 nM primers and 7 μl of RNase-free water (Qiagen). The primer sequences and PCR amplicon sizes are summarized in Table 2.

Primers previously designed for the insect *COXI* gene (*Simon et al., 1994*; *Tanaka et al., 2001*) were used to amplify a 630-base pair (bp) fragment from the 3′ end of the locus. The amplification conditions comprised an initial denaturing step at 95 °C for 5 min followed by 35 cycles of denaturing at 94 °C for 30 s, annealing at 52 °C for 40 s, and extension at 72 °C for 45 s, and a final extension phase at 72 °C for 5 min. For the mitochondrial *16S* gene, the scorpion-specific primer pairs modified by *Gantenbein et al. (2005)* were used to amplify a 425-bp region at the 3′ end of the locus. The amplification conditions comprised an initial denaturing step at 94 °C for 4 min followed by 30 cycles of denaturing at 94 °C for 30 s, annealing at 47.5 °C for 30 s, and extension at 72 °C for 30 s, and a final extension phase at 72 °C for 7 min. The *COXI* and *16S* gene fragments were also amplified from three specimens keyed out as *Urodacus manicatus* (Um2714, Um1814) and *U. novaehollandiae*

**Table 2  Primer and amplicon details.** List of primer sequences and corresponding amplicons sizes for the three *Urodacus yaschenkoi* loci (*COXI*, *16S* rRNA, *28S* rRNA).

| Marker | Primer | Primer sequence | Size (bp) | Reference |
|---|---|---|---|---|
| *COXI* | F C1-J-2183 | 5′-CAACATTTATTTTGATTTTTTGG - 3′ | 550–630 | *Simon et al. (1994)* |
|  | R COXIKG-R2 | 5′- GATATTAATCCTAAAAAATGTTGAGG-3′ |  | *Tanaka et al. (2001)* |
| *COXI* | Nested F | 5′-AGGAACCTTTTGGGGCTTT-3′ | 150 |  |
| *COXI* | Nested R | 5′-AGGAACCTTTTGGGGCTTT-3′ |  |  |
| *16S* | F 16SF | 5′- AACAAAACCCACAGCTCACA- 3′ | 422 | *Gantenbein et al. (2005)* |
|  | R 16SR | 5′- GTGCAAAGGTAGCATAATCA- 3′ |  |  |
| *28S* | R1 | F R1S (5′-ACCCGCTGAATTTAAGCAT-3′), R R1AS (5′- GCTATCCTGAGGGAAACTTC-3′) | 1,158 | *Arabi et al. (2012)* |
|  | R2 | F R2S (5′-CGACCCGTCTTGAAACACGGA-3′), R R2AS (5′-CACCTTGGAGACCTGCTGCGGAT-3′) | 1,246 |  |

(Un2112, Table 1). Sequences from these taxa were used as outgroups in downstream phylogenetic reconstruction. Primer pairs R1S and R1AS, and R2S and R2AS, designed by *Arabi et al. (2012)*, were used to amplify 1,158-bp and 1,246-bp fragments of the *28S* locus, respectively. Each set of primers amplifies a different region of the gene, which overlaps by 327 bp, and their sequences were concatenated to form a larger product of 2,076 bp. The amplification conditions for both sets of primers comprised an initial denaturing step at 94 °C for 4 min, followed by 30 cycles of denaturing at 94 °C for 30 s, annealing at 55 °C for 30 s, and extension at 72 °C for 30 s, and a final extension phase at 72 °C for 7 min.

Museum specimens that were not stored under ideal conditions for preservation failed to yield *COXI* amplicons suitable for direct sequencing. To address this issue, additional PCR primers were designed to amplify smaller fragments for *COXI* locus (Table 2), resulting in amplicons of 150 bp that were used for subsequent analysis. For the SAM specimens, the amplification of the *28S* nuclear gene failed entirely and these samples were excluded from further analysis of the nuclear gene variation. All amplicons were sequenced in both directions using the PCR amplification primers, and carried out on an Applied Biosystems 3130 genetic analyzer by Macrogen Inc. (Seoul, South Korea).

Sequences were aligned and edited in Geneious Pro v6.1 (Biomatters Ltd.) using the MUSCLE alignment option with default parameters. All chromatograms were checked for the presence of multiple peaks (which indicate heterozygosity), and authenticity of the *COX1* coding gene was validated by checking for indels and premature stop codons. After this editing process, the alignment of the mitochondrial gene fragments yielded 616-bp and 396-bp products for the *COX1* and *16S* genes respectively, and the final *28S* alignment was 2,076 bp in length. The final dataset contained 68 sequences for each of the mitochondrial genes and 27 sequences for the *28S* locus (Table 1, (GenBank accession # KP176717–KP176786)). Shared haplotypes were identified and the uncorrected pairwise genetic distances (%) were calculated using Geneious Pro v6.1 (Biomatters Ltd.). This simple distance measure was implemented to achieve reliable estimates of both intraspecific and interspecific genetic variation.

## Phylogenetic analysis

Phylogenetic reconstructions and divergence dates among lineages were calculated using unique haplotypes and Bayesian Inference (BI) methods implemented in BEAST v2.1.3 (*Bouckaert et al., 2014*). We used jModeltest v0.1.1 (*Posada, 2008*) to select the best-fit model of evolution, based on Akaike Information Criteria (AIC) (*Akaike & Company, 1981*) for each of the mitochondrial and nuclear genes (GTR + G in each case). Mitochondrial loci were combined for analysis due to their similar modes of evolution (GTR + R), as indicated by the incongruence-length difference (ILD) tests (*Farris et al., 1995*) implemented in PAUP_4.0b10 (*Swofford, 2002*). The nuclear gene (*28S*) was analyzed independently due to inconsistencies in taxon sampling (Table 1).

Operators were auto-optimized, and five independent Markov Chain Monte Carlo (MCMC) runs were performed using a Yule (speciation) tree-prior, each running for 5 $\times 10^6$ generations, sampling every 10,000 states. Log files were examined with Tracer v1.5 (*Rambaut et al., 2014*) to ensure that runs were sampling from the same posterior distribution, to determine appropriate burn-in, and to ensure that effective sample sizes (ESSs) of parameters of interest were greater than 1,000. Tree files of independent runs were then combined using LogCombiner v2.1.3 (*Drummond et al., 2012*), discarding the first 20% and re-sampling at a lower frequency of 15,000. The maximum clade credibility (MCC) tree was recovered from a sample of 10,000 posterior trees, and branch support was annotated using TreeAnnotator v2.1.3 (*Drummond et al., 2012*). Each analysis started with a random starting tree and seed with no root specified. Sequence data from species of the same genus (*U. manicatus* and *U. novaehollandiae*) were used to estimate the root of the mitochondrial gene tree.

Additional phylogenetic constructions were also performed using a truncated *COXI* alignment to test the influence of missing data on the final tree topology. Because numerous museum collections yielded short *COXI* gene products, we trimmed the alignment to 150-bp to exclude regions of the alignment with high levels of missing data. This exercise demonstrated that the inclusion/exclusion of missing data had little influence on the phylogenetic reconstructions. Consequently, all results presented from this point reflect those from the non-truncated *COX1* alignment.

## Species delineation based on molecular data

We implemented two DNA taxonomy approaches to evaluate the presence of cryptic species. First, the general mixed Yule coalescent (GMYC) approach (*Fujisawa & Barraclough, 2013*; *Pons et al., 2006*) was applied to an ultrametric tree (produced using BEAST) in *R Development Core Team (2008)* with the Splits package (http://splits.r-forge.r-project.org). The GMYC model is a process-based approach that detects the threshold in a gene tree at which within-species processes (i.e., coalescence) shift to between-species processes (i.e., speciation and extinction). Second, we combined the Poisson Tree Processes model for species delimitation (PTP) and a Bayesian implementation of PTP (bPTP) to infer putative species boundaries on a given phylogenetic input tree (*Zhang et al., 2013*). The PTP/bPTP model, unlike the GMYC model, requires a bifurcated phylogenetic tree rather than an ultrametric tree. PTP/dPTP models speciation or branching events in terms of the number of substitutions. The following parameters were used: MCMC, 500,000

generations; thinning, 100; burn-in, 0.1; seed, 123, and assessed convergence in each case to ensure the reliability of the results.

## Delineation based on the analyses of morphological measurements

Proportions of several characters that approximate body shape were assessed in 39 female adult specimens that were keyed out as *U. yaschenkoi* (according to *Koch, 1977*) and were collected at 26 locations (Table 1, Fig. 1). Gender was determined by examining the genital opercula of adult scorpions, with males having a small finger-like projection known as the genital papilla. Because our collection contained only three males, the analyses were done only with females.

The following traits were measured under a microscope using an ocular ruler with 1-mm precision: carapace length (CL), metasoma segment V length (MVL), telson length (SL), pedipalp length (PL), chela length (ChL), pecten length (PecL) and pecten width (PecW). Ratios of traits (e.g., CL/MVL, SL/PL etc.) gave in total 21 variables scored in each individual (Supplemental File 4). These variables were treated as predictors in the Linear Discriminant Analysis (LDA) implemented in the R package "MASS" (*Venables & Ripley, 2002*). LDA was used to test whether the linear combination of 21 variables (ratios of morphological measurements) can predict individual's membership to a mitochondrial lineage (putative species). Strong predictive power of morphological variation on the observed molecular divergence would provide additional support for a species complex in *U. yaschenkoi*.

## Divergence time estimation

The mitochondrial gene tree was time calibrated with divergence times of nodes inferred from 95% highest posterior density (HPD) intervals. Scorpion-specific mutation rates of 0.007 substitutions/site/million years for *COXI* and 0.005 substitutions/site/million years for *16S* (*Gantenbein et al., 2005*; *Gantenbein & Largiadèr, 2003*) were used to calibrate the tree. These estimates are derived from buthid scorpions and have been used to estimate divergence times among various scorpion lineages including non-buthid taxa (*Bryson et al., 2013a*; *Bryson et al., 2013b*; *Graham, Oláh-Hemmings & Fet, 2012*). Substitution rates were set in BEAUti v1.7.3 (*Drummond et al., 2012*) using relaxed clock log normal priors. Tracer was then used to obtain parameter estimates for time to most recent common ancestor (tMRCAs) for nodes within the gene tree.

## RESULTS

We identified 31 unique mitochondrial haplotypes with uncorrected distances between haplotypes ranging from 0.3–7.6% (mean ± standard deviation = 3.0% ± 0.4%) and distances from the outgroup taxa of 8.4–10.2% (mean ± standard deviation = 9.4% ± 1.4%) (Supplemental File 1). A total of 13 nuclear *28S* haplotypes were identified with uncorrected *p*-distances of 0.1–0.5% (mean ± standard deviation = 0.2% ± 0.1%) (Supplemental File 2). A list of haplotypes for sample locations is provided in Supplemental File 3.

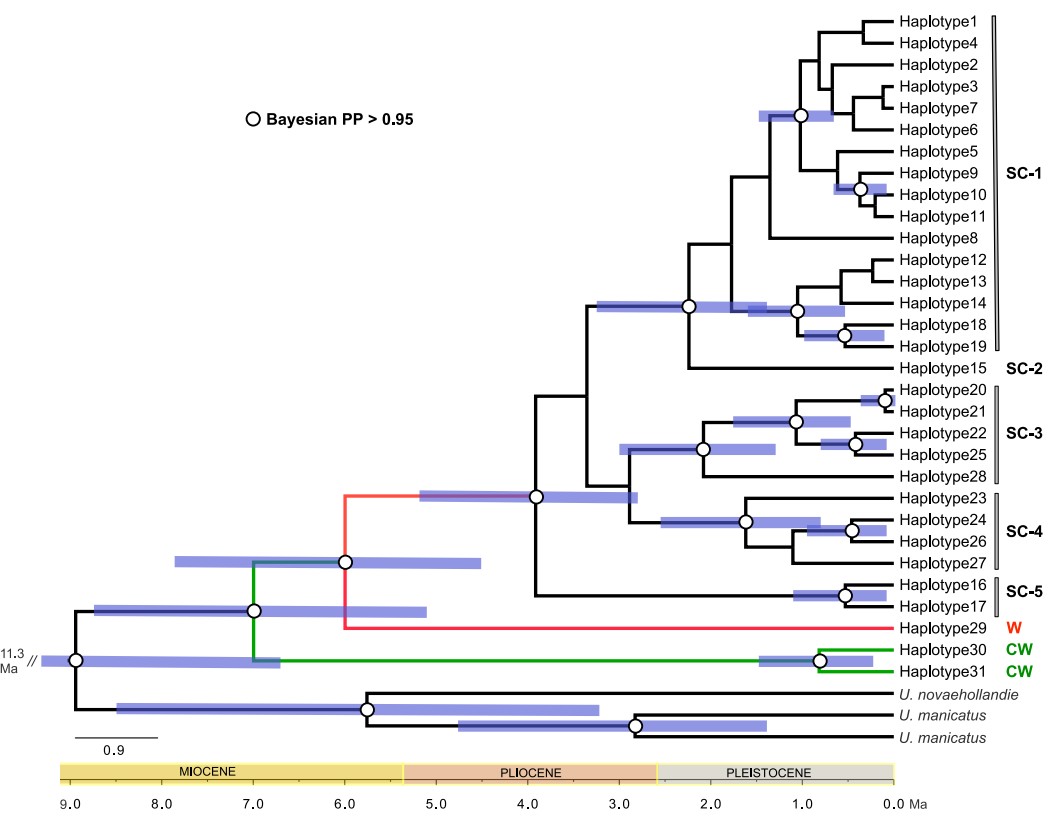

**Figure 2** **Dated phylogeny (Bayesian tree) for *Urodacus yaschenkoi* based on the concatenated *COXI* and *16S* partial sequences.** Putative species inferred with the PTP/bPTP approach are marked as SC1-5, CW and W. The 95% CI for each divergence time is shown in blue.

## Phylogenetic analysis
### Mitochondrial markers

Bayesian inference analysis of the mitochondrial dataset identified several genetically divergent lineages (three major lineages represented as black, red and green clades in Fig. 2), with strong statistical support for their respective monophyly (posterior probability > 0.95). Sublineages within the black clade are broadly distributed across Victoria, South Australia and Western Australia, whereas the red and green clades are restricted to Western Australia (Fig. 1). From this point forward we will refer to the black, red and green clades as the south-central (SC), western (W) and central-western (CW) lineages, respectively.

Mean uncorrected pairwise genetic distances between the three major lineages (SC, CW and W) ranged from 6.4 to 6.9 (overall mean ± standard deviation = 6.6% ± 0.9%). The mean sub-lineage distances ranged from 2.2% ± 0.4% and 0.8% ± 0.2%, respectively (not calculated for the W lineage due to only a single recorded haplotype). Mean uncorrected distances between the three major lineages and the outgroups ranged from 9.3 to 10.3% (mean ± standard deviation = 9.4% ± 1.4%).

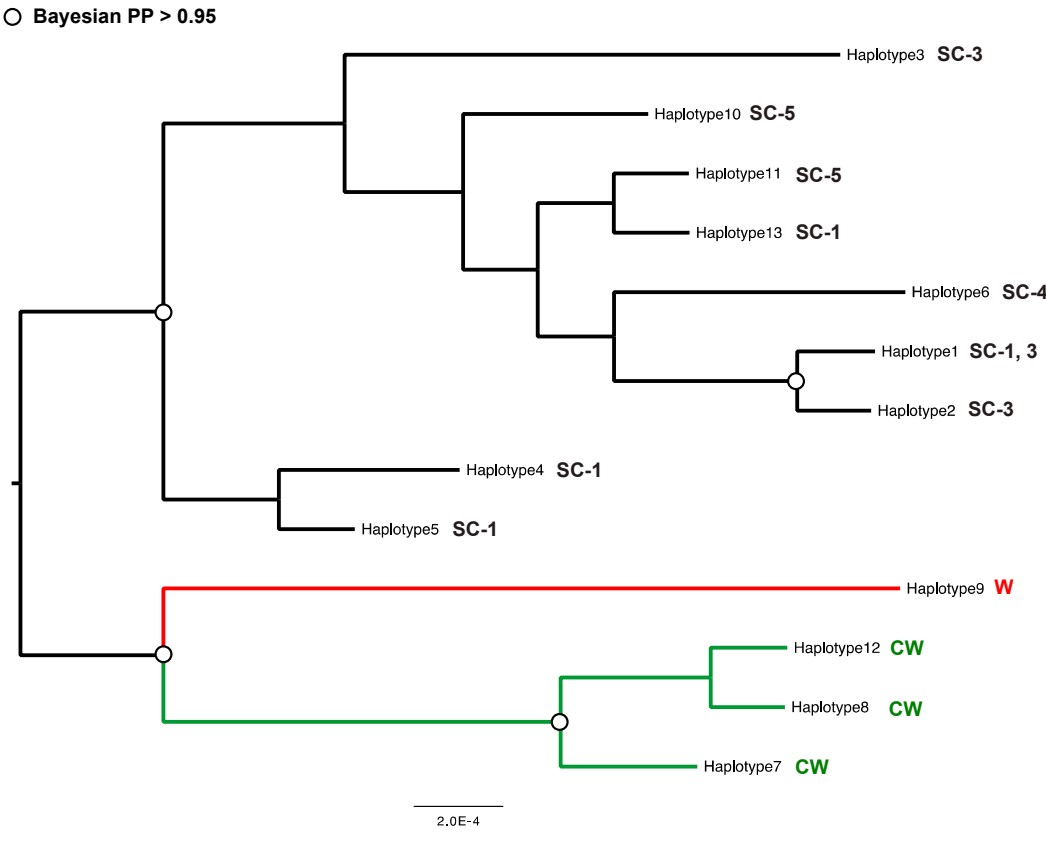

○ Bayesian PP > 0.95

Haplotype3 **SC-3**
Haplotype10 **SC-5**
Haplotype11 **SC-5**
Haplotype13 **SC-1**
Haplotype6 **SC-4**
Haplotype1 **SC-1, 3**
Haplotype2 **SC-3**
Haplotype4 **SC-1**
Haplotype5 **SC-1**
Haplotype9 **W**
Haplotype12 **CW**
Haplotype8 **CW**
Haplotype7 **CW**

2.0E-4

**Figure 3** Bayesian unrooted tree for *Urodacus yaschenkoi* based on the *28S* partial sequences.

### Nuclear marker

Despite low level of variation in the *28S* dataset, Bayesian analysis produced a nuclear gene topology that was largely concordant with the mitochondrial gene tree. Three genetically divergent clades were identified, corresponding to those from the mitochondrial dataset (SC, CW and W, Fig. 3). In each case, strong statistical support for the monophyly of each clade was found (posterior probability > 0.95). The unresolved interrelationships among lineages within each clade in the nuclear gene tree prevented any reliable inferences of phylogeographic patterns.

### Molecular-based species delineation

Among the 31 unique mitochondrial haplotypes described above, the GMYC model identified nine entities and the PTP/bPTP approach identified seven, each representing putative species (Table 3). The assignment of haplotypes to putative species groups is shown in Fig. 2, where conspecifics share a common number. Species assignments were highly consistent when comparing each of the methods, but we presented the PTP/bPTP results as they are more accurate when the evolutionary distances between lineages are small (Zhang et al., 2013). In summary, SC, W and CW clades were recognized as putative species groups, as were the sub-lineages within the SC ancestral grouping (SC-1 to 5, Fig. 2).

**Table 3** **Molecular species delineation analyses.** Species delineation analyses in *Urodacus yaschenkoi* based on 31 unique mitochondrial haplotypes.

| Analysis type | # Entities | Statistics |
|---|---|---|
| GMYC | 9 | Likelihood null model: 32.7519; likelihood best model: 33.36569; likelihood ratio: 1.2255; *P*-value, 0.0001, confidence interval: 1–10 |
| PTP/bPTP (ML and BL) | 7 | Acceptance rate: 0.50975; merge: 49,942; split: 50,058 |

## Discriminant power of morphological variation

None of the *U. yaschenkoi* specimens that were characterized at 21 morphological ratio variables were assigned to the W mitochondrial clade, hence the LDA was done on 39 females assigned to the SC and the CW clades. Individuals were categorized into four groups (putative species) based on the results of the PTP/bPTP molecular species delineation analysis: 18 females from SC-1, 12 from SC-3, three from the SC-4, and six from the CW clade (Fig. 2). Because our dataset contained four groups, we could find a maximum of three discriminant functions that separate these groups.

The first discriminant function (LD1) achieved 93.7% of the separation, reflecting the morphological distinction of the CW clade from the SC clade (Fig. 4). Further separation of the three putative groups within the SC clade was weak (LD2-3, Fig. 4). We then grouped samples into two putative species (CW and SC clade) and tested the accuracy of prediction using 100 jackknife resampling steps. The grouping into two molecular clades based on morphological variation was 100% accurate (33/33) for the SC clade and 83.3% accurate (5/6) for the CW clade. Therefore, our results indicate strong predictive power of body proportion variation on the observed molecular divergence, and suggest the existence of at least two distinct taxa within *U. yaschenkoi.*

The most discriminating uncorrelated proportions were of the telson and chela length (SL/ChL) and pedipalp and pecten length (PL/PecL). Overall, members of the CW clade tend to have disproportionately shortened chela and enlarged pecten when compared to the members of the SC clade.

## Divergence dating

Our time calibrated mitochondrial phylogeny suggested that the split between the major *U. yaschenkoi* clades (SC, CW and W lineages) occurred during the late Miocene/early Pliocene (4–7 MYA) (Fig. 2). Lineage diversifications within SC appear to have occurred during the Pliocene and early Pleistocene (1.8–4 MYA), while finer-scale phylogeographic patterning within the sub-lineages arose during the late Pleistocene (<1 Mya). Divergence time estimates should be interpreted with some caution, as the nucleotide substitution rate was derived from a different scorpion family (Buthidae) and there are large errors margins around 95% HPD estimates.

## Biogeographic patterns

The SC lineage showed substantial geographic structure. The most divergent sub-lineage (SC-5) was found in Western Australia in sympatry with the CW lineage (Fig. 1). SC-1 was found west of the Central Ranges, through to the Eyre Peninsula in South Australia, while

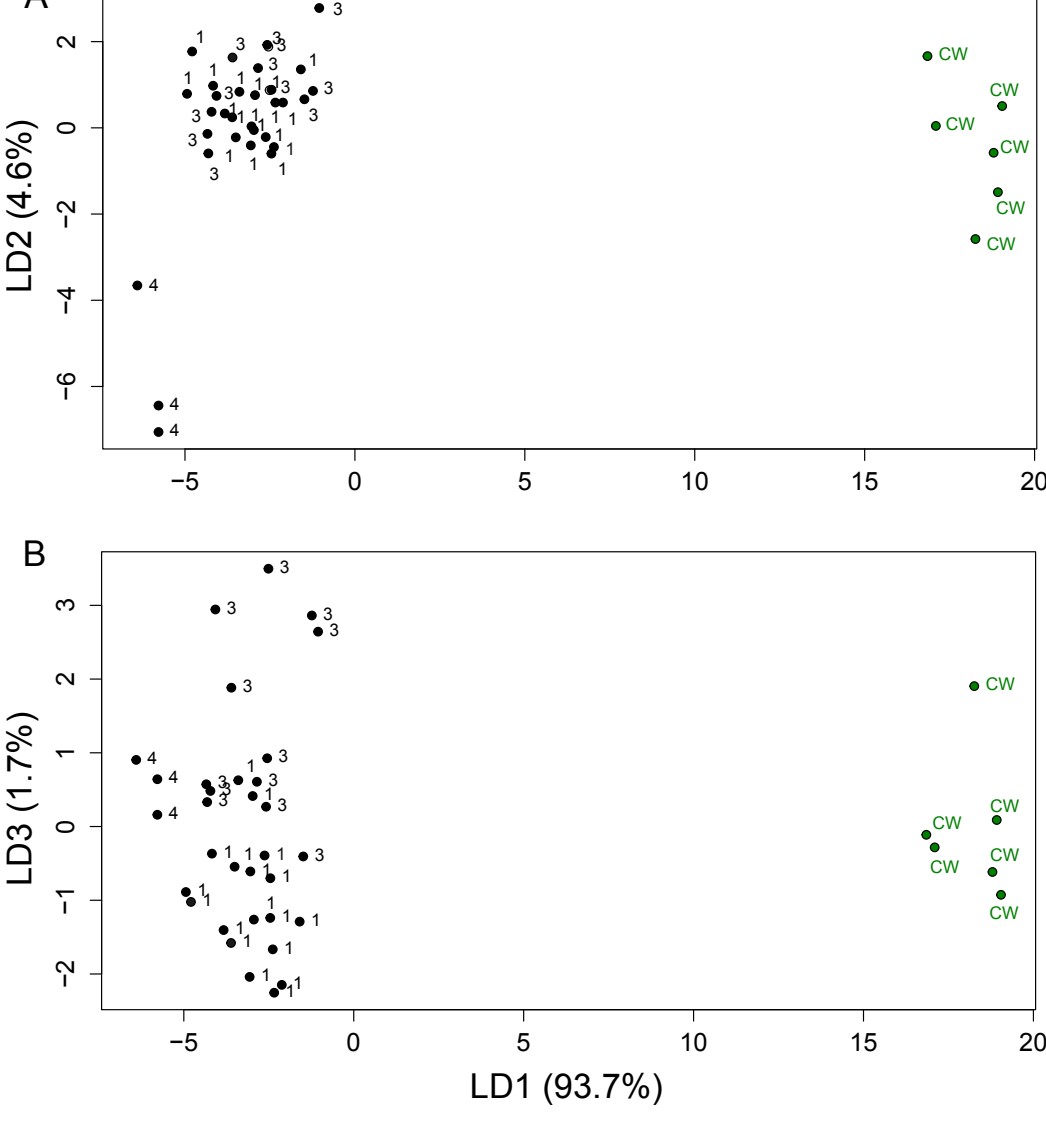

**Figure 4  LDA for body proportions.** Individual scores for the first 3 axes of Linear Discriminant Analysis. 21 body-proportions were measure in Urodacus yaschenkoi adult females. Numbers (1, 3, 4) denote individuals belonging to one of the SC sublineages (SC1, 3, 4), and CW denotes individuals from the CW clade.

SC-3 had a distribution extending from the Central to Mt Lofty Ranges in South Australia, and across to north-western Victoria. SC-4 had a narrow north-south distribution in the central inland and coastal regions of South Australia (Fig. 1).

## DISCUSSION

Our analyses reveal strong genetic and morphological diversification in *U. yaschenkoi* across its range, pointing to the existence of a species complex with at least three putative species High statistical support for the monophyly and the extent of genetic divergence between the main three lineages (6.4–6.9%) exceeds estimates of interspecific divergence previously

reported for other scorpion and arthropod groups (*Bryson et al., 2014*; *Tourinho, Sole-Cava & Lazoski, 2012*; *Wysocka et al., 2011*). DNA-based species delineation approaches (GMYC and bPTP) provided significant statistical support for the recognition of the three lineages (SC, CW, W) as distinct species, and potential further cryptic speciation within the south-central clade (SC1-5, Fig. 2).

We also demonstrated a strong association between this molecular divergence and morphological variation. Namely, ratios of several traits that approximate body shape had a strong predictive power (83–100%) in discriminating two major molecular clades (CW and SC). The two clades differ most notably in proportions involving chela and pecten. Because of their great variation in shape, scorpion chalae have been used as one of the key characters to delineate different ecomorphotypes (*Van der Meijden, Kleinteich & Coelho, 2012*). Until now *U. yaschenkoi* has been distinguished from other congeneric species by its much smaller terminal prolateral tarsal ungues and by the production of large amounts of venom (*Koch, 1977*). Based on our results from a limited sample size, detailed analyses of morphological variation in *U. yaschekoi* are warranted.

Our time-calibrated phylogeny suggests that the split between the CW, W and SC clades occurred during the mid-Miocene to early Pliocene (approximately 5–9 Mya). This geological time was marked by a shift to a much drier climate, the significant contraction of rainforests and the expansion of arid habitats (*Martin, 2006*). Further diversification within the major ancestral *U. yaschenkoi* lineages appears to have occurred throughout the Pliocene (3–5 Mya), which was a consistently dry period. This is followed by further lineage divergence during the mid and late Pleistocene when the climate was highly dynamic (<1 Mya), with wetter and drier episodes corresponding to interglacial and glacial cycles (*McLaren & Wallace, 2010*).

The spatio-temporal dynamics of diversification observed in *U. yaschenkoi* parallels those reported in other Australian arid biota. Reviewing tens of dated phylogenies of the south-western Australian terrestrial fauna, including arthropods like crayfish and spiders, *Rix et al. (2015)* found a compelling commonality in the basal east–west lineage diversification during the first half Miocene (until 10 Mya). The more xeric taxa currently occupying semi-arid and arid zones seemed to have experienced this divergence in late Miocene (6–10 Mya) (*Rix et al., 2015*), which we also inferred in the desert scorpion *U. yaschenkoi* (Fig. 2). A strong genetic and morphological divergence between the *U. yaschenkoi* lineages from the western (CW, W) and south-central (SC) Australia could be partly explained by the Miocene east–west vicariance hypothesis (*Rix et al., 2015*) (Fig. 1). After a longer period of range contraction, arid-adapted taxa such as *U. yaschenkoi* likely underwent significant range expansions during the Pliocene. Separation of SC-5 from other SC sub-lineages was estimated to have occurred during this time (Fig. 2), with SC-5 moving easterly. This sub-lineage is now sympatric with the CW clade (Fig. 1), suggesting their secondary contact. Further diversification within the SC clade (SC1-4) coincides with transition to the Pleistocene severe glacial cycles and expansion of the Australian deserts during the last 1 My (beginning of the "dusty world", *Rix et al., 2015*). Like theBynoe's gecko (*Fujita et al., 2010*) and lizards (*Dubey & Shine, 2010*; *Pepper et al., 2011*), *U. yaschenkoi* is another arid-adapted Australian taxon whose diversification and distribution were profoundly

affected by the opening of desert biomes during this hyper-arid, unstable climatic history. Teasing out the relative importance of vicariance, putative refugia (e.g., Pilbara, Kimberley, central Ranges, (*Pepper, Doughty & Keogh, 2013*), or dispersal (*Melville et al., 2016*) (Fig. 1) in shaping this diversity would require extensive sampling, particularly at the western and northern parts of *U. yaschenkoi* distribution.

### Revising the *U. yaschenkoi* taxonomy—future directions

Our results provide solid baseline data on the historical and spatial extent of diversification in *U. yaschenkoi* and offer some guidelines for future integrative taxonomic approaches in delimiting species within this taxon. We found an agreement among disciplines (morphology, nuclear and mitochondrial genetic information) during a primary exploration, which strengthens the argument for a taxonomic revision (*Pante, Schoelinck & Puillandre, 2014*; *Schlick-Steiner et al., 2009*). Congruent morphological and molecular phylogenetic signals are particularly compelling for a scorpion taxon, given that this is not the case in many scorpion lineages (*Sharma et al., 2015*).

The level of mitochondrial sequence divergence observed between *U. yaschenkoi* lineages satisfy the requirements for species delineation based on the principles of the phylogenetic species concept (*De Queiroz, 2007*; *Wheeler, 1999*), The three major lineages (SC, CW, W) can be considered the putative species. Because genetic 'yardstick' approaches provide crude taxonomic measures and nucleotide substitution rates often vary considerably between taxonomic groups, some caution is needed when considering findings of these analyses alone. Additional DNA-based species delineation approaches (GMYC and bPTP) indicated extensive cryptic speciation in *U. yaschenkoi* (Fig. 2). The GMYC method has been criticized for over-splitting species with a pronounced genetic structure (*Satler, Carstens & Hedin, 2013*), yet several recent studies have shown that it is highly robust (*Fujisawa & Barraclough, 2013*; *Talavera et al., 2013*) The obvious next step is to characterize the nuclear genome-wide variation in *U. yaschenkoi* sampled extensively within the "type" locality (28°35′S, 138°33′E), as well as western and northern parts of the distribution. We certainly advise against a pool-sequencing phylogenomic approach (e.g., samples from the same location are pooled to achieve cost-efficiency), given that the putative species have been found in sympatry.

The proportions of various morphological characters are routinely used in species descriptions or identification keys, particularly for arthropods where morphologically similar species often differ significantly in body proportions but not in qualitative characters (*Baur & Leuenberger, 2011*). Arguably, the results of multivariate analyses summarizing the overall body shape differences between groups are not easily interpreted. Yet, our initial results suggest that further analyses of e.g., chela shape might reveal more easily quantifiable diagnostic characters for *U. yaschenkoi*. Several parameters of chala shape were found to be correlated with the amount of strain stress they can withstand. Specifically, slender chela morphologies may be less suitable for high-force functions such as burrowing and defence (*Van der Meijden, Kleinteich & Coelho, 2012*). Given that *U. yaschekoi* putative species (SC and CW) show marked shape differences involving chela, further exploration of burrowing behavior or pray preference might provide additional characters to describe the *U. yaschenkoi* species complex.

Finally, it is important to note that we cannot exclude the possibility that some of the cryptic lineages have already been described as species, and we are not able to compare our genetic data against other *Urodacus* sequences as none published at the time of our study. Also, our sampling did not cover the exact "type" locality (28°35′S, 138°33′E). The samples closest to this area belong to the SC clade and likely represent the "type" lineage These data gaps would need to be addressed in further studies aiming to revise the taxonomy of the Australian desert scorpion *U. yaschenkoi*.

## CONCLUSIONS

Our study provides the first insight into the molecular phylogeny of the endemic Australian scorpion *Urodacus yaschenkoi*. We show that this scorpion shares a complex diversification history with other Australian arid-adapted fauna. Concordance between the mitochondrial and nuclear data, along with the morphological variation, all suggest that *U. yaschenkoi* is a species complex that requires further taxonomic revision. Our findings highlight the importance of conserving populations from different Australian arid zones in order to preserve patterns of endemism and evolutionary potential.

### Funding

KLR was supported by scholarships from CONACyT and from The Hugh Williamson Foundation, through Museum Victoria. The funders had no role in study design, data collection and analysis, decision to publish, or preparation of the manuscript.

### Grant Disclosures

The following grant information was disclosed by the authors:
CONACyT.
The Hugh Williamson Foundation.

### Competing Interests

The authors declare there are no competing interests.

### Author Contributions

- Karen Luna-Ramirez conceived and designed the experiments, performed the experiments, contributed reagents/materials/analysis tools, wrote the paper.
- Adam D. Miller and Gordana Rašić analyzed the data, contributed reagents/materials/analysis tools, wrote the paper, prepared figures and/or tables, reviewed drafts of the paper.

### DNA Deposition

The following information was supplied regarding the deposition of DNA sequences:
The sequences described here are accessible via GenBank accession numbers KP176717–KP176786.

## Data Availability

The raw data has been supplied as Supplementary File.

## Supplemental Information

Supplemental information for this article can be found online at http://dx.doi.org/10.7717/peerj.2759#supplemental-information.

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
