# Peer review of "Genetic and morphological analyses indicate that the Australian endemic scorpion Urodacus yaschenkoi (Scorpiones: Urodacidae) is a species complex"

_PeerJ, doi:10.7717/peerj.2759_

## Round 0.1 · original submission · Minor Revisions

The three reviewers all see merit in your paper. However, they all have some suggestions as to improvements that would be necessary before it was acceptable. In particular, I agree with reviewer 1, that that the current justification for the study is very focused on the medical importance, whereas most of the approach is more tailored towards a biogeographic study. Therefore I recommend that the introduction is refocused to reflect the real findings of the paper. the medical implications can still be mentioned, but if we only want to know which clades are species, many of the analyses are not really relevant.

Also, further details about the specimens should be provided and, as suggested by two reviewers, the clades should be given simple names so that discussion around them becomes easier.

Reviewer 1 ·

Basic reporting

The main premise of the work is very focused on the difficulty of doing medical research with U. yaschenkoi if it actually represents many species. Can you cite studies where this may have been a problem? Where do these studies source specimens? Is it likely that they are falling into this error and undermining their findings? If there is no real evidence that this is an actual problem in the medical world, then the Intro/Discussion should probably be restructured to be about biogeography rather than cryptic species in medical research.

The Genbank accession numbers need to be reported

In your results you describe the distribution of a number of lineages, without explaining the basis for recognizing these lineages (lines 269-284). Then, in a later section you describe a method for recognizing these lineages (lines 311-319). I would recommend delineating the lineages first, then describing their geographic patterns after. When you delineate the lineages, number them and use this numbering system to refer to them for the rest of the manuscript. At the moment the shift from colours, to regions to shapes makes it very hard to follow the thread of the results/discussion.

Experimental design

You can easily integrate your 3 genes by building a partitioned gene tree in RaxML or MrBayes. It would be useful to do this and present a single gene tree. Then present your time tree.

Does the BSP analysis offer anything to your study? I do not think it does and you should probably remove it.

Will you be taking these results and describing your putative new species? If not, you should probably provide the information in this study to allow a taxonomist to follow this work up and describe these new species. Even if you are intending to, I recommend adding this info to make this paper integrate with any future publications you do. The information you should add is; the unregistered specimens should be lodged at a museum and their reg/ID numbers presented in Table 1, Table 1 should report all genbank #s against their respective specimens, and specimens need to be linked to your lineages. At the moment it would be extremely had to recreate this study with the information provided. Also, can you guess at what "true" U. yaschenkoi is? What lineage would keep the name U. yaschenkoi.

Can you exclude the possibility that any of the cryptic lineages are in fact another, described species? You have attempted to do this morphologically, but can you use the genetic data to compare to any sequences that exist elsewhere. If you cannot exclude this possibility, you need to raise it, as the morphological traits may not be very reliable.

Validity of the findings

The validity of the findings are sound.

Additional comments

Minor comments:
Title: Add “morphology” to your title, as you didn’t rely solely on genetic data.
Line 82: 20 described species, with many likely undescribed species
Line 142-145: These specimens are not in Table 1
Line 155-156: This is a little misleading, as you still included them in your analysis for other genes.
Line 336-337: Why were they not included? Because they were males?
Line 371: The methods were not consistent.

·

Basic reporting

No comments

Experimental design

No comments

Validity of the findings

No comments

Additional comments

This manuscript assesses population structure within an Australian scorpion species, U. yaschenkoi. The authors collate a solid dataset of multilocus molecular sequence and morphological characters and analyse the data appropriately to answer the questions asked. The data supports multiple genetic lineages within the species, with some that are also associated with morphological differences. The manuscript is well-written throughout, clearly argued and the figures and tables represent the data well. I have some relatively minor comments below, and some mostly textual suggestions on the attached annotated pdf, but have no trouble recommending this manuscript for publication if these comments are addressed.

Minor comments:

The authors use a lot of circumspection in the discussion around exactly how many ‘species’ are supported in the data. I think I’d like to see something more specific on this from them, rather than just that the data supports U. yaschenkoi as a species complex. I think the authors could say with some certainty that the two groups supported by the morphological data (south-central & western-central) are most likely to be species-level units, and equally the western lineage (although this remains untested morphologically). The authors have all the caveats and caution, but I’d like to see the case argued for how many ‘species’ the data supports.

I think the authors need to consider a coding/reference system for their sublineages because current descriptions are quite wordy (describing based on the symbol used in the figure). I think the authors could use, for example, SC-1 for south-central sublineage 1, or similar. This would free up some words in the results especially and make it more clear.

I wonder why the authors haven’t analysed the morphological data more fully, assessing individual characters for significant differences between groups, etc. They could also consider scoring them for use in a phylogenetic analysis in MrBayes for example. It would be interesting to know whether a morphological tree matched the molecular tree, even with less samples.

The authors really should cite more of the mainstream integrative taxonomy literature, especially from the insect literature (eg., Schlick-Steiner et al, 2010, Ann Rev Ent.; Schutze et al, 2016, Ann Rev Ent).

The end of the discussion could use some more detail on exactly what future work should focus on. This feeds into my major comment above because the ideas/interpretation of how many species there are will in some way determine what future research needs to be done, because of the gaps left from the current study.

·

Basic reporting

This manuscript tests the species status of Urodacus yaschenkoi, using molecular and morphological data. It is very well written, with few grammatical errors. I especially like the way the authors have written the discussion with the central aim in mind. As far as I can ascertain, this manuscript meets PeerJ standards throughout, and I have further no comments with respect to basic reporting.

Experimental design

The experimental design is for the most part sound, and appropriately tailored to the aims of the study. However, I have two minor issues.

Firstly, the authors are testing the species status of U. yaschenkoi, but at no point do they discuss the type locality of this species. For these sorts of studies, it is important that this is acknowledged and that type localites are preferably sampled specifically. The type locality is Killalpaninna, Coopers Creek, South Australia [28°35’S, 138°33’E]. It appears to me as though no specimens were collected from this area, although the north-eastern records to the east of the Central Ranges are not too far away. While I accept that based on these data the black clade is probably conspecific with specimens from Cooper Creek, this needs to be carefully argued when a type locality is not sampled.

Secondly, it is a shame the trees were not rooted on scorpions from other families. Even when trees are analysed unrooted, when congeners are used as outgroups it assumes monophyly of the focal taxon. Phylogenetic analyses always benefit from rigorous outgroup sampling, and given the multitude of sequences on GenBank, two other taxa could also have been used in addition to the Urodacus.

Validity of the findings

Overall I think the findings are valid and carefully discussed. Indeed, I think the authors have done an excellent job at interpreting their results, However, I do question the use of the refugia of Koch (1977) in Figure 1 and the primary use of this reference in the text (line 395 onwards). These refugia hypotheses are highly simplistic, and do not take in to account the large amount of information we now have for the arid zone. I suggest the authors consult Pepper et al. (2013: Pilbara), Rix et al. (2015: SW. W.A.) and the multitude of papers that have come out of the work of Craig Moritz and colleagues (herpetofauna), and Mark Harvey and colleagues (arachnids), which explore more refined and complex refugia hypotheses in both the arid and mesic zones. I realise these are points of discussion, more than anything, but relevant nonetheless given the potential speciation of this lineage in WA.

References:
Pepper, M., Doughty, P. and Keogh, J.S. (2013). Geodiversity and endemism in the iconic Australian Pilbara region: a review of landscape evolution and biotic response in an ancient refugium. Journal of Biogeography 40: 1225-1239.

Rix, M.G., Edwards, D.L., Byrne, M., Harvey, M.S., Joseph, L. and Roberts, J.D. (2015). Biogeography and speciation of terrestrial fauna in the south-western Australian biodiversity hotspot. Biological Reviews 90: 762-793.

Additional comments

I have few other comments. These are outlined below.

- In the first paragraph, I think it is inappropriate to discuss scorpions as "among the most ancient arthropods" and as "living fossils". These quaint concepts are hardly scientific. Most arachnid orders appeared in the Devonian, and recent research has shown that crown group Scorpiones may be much more recent in age. Similarly, all Recent taxa are as 'modern' as you and I. And at some level, we're all living fossils. This is pretty meaningless. It would be better to simply state the ancient Paleozoic origins of the order, and their largely unchanged Bauplan over evolutionary time without reference to ‘living fossils’.

- Lines 197-198: "Buthid" should be "buthid". Family-group names are not capitalised when used as adjectives.

- Line 271: "monophylies" should be "monophyly"

Respectfully submitted,
Mike Rix

---

## Round 0.2 · accepted · Accept

I think the paper is now much improved in focus and is now acceptable for publication.